# An Influenza A Vaccine Based on the Extracellular Domain of Matrix 2 Protein Protects BALB/C Mice Against H1N1 and H3N2

**DOI:** 10.3390/vaccines7030091

**Published:** 2019-08-19

**Authors:** Hui Kian Ong, Chean Yeah Yong, Wen Siang Tan, Swee Keong Yeap, Abdul Rahman Omar, Mariatulqabtiah Abdul Razak, Kok Lian Ho

**Affiliations:** 1Department of Pathology, Faculty of Medicine and Health Sciences, Universiti Putra Malaysia, UPM Serdang 43400, Selangor, Malaysia; 2Department of Microbiology, Faculty of Biotechnology and Biomolecular Sciences, Universiti Putra Malaysia, UPM Serdang 43400, Selangor, Malaysia; 3Institute of Bioscience, Universiti Putra Malaysia, UPM Serdang 43400, Selangor, Malaysia; 4Department of Marine Biotechnology, China-ASEAN College of Marine Sciences, Xiamen University Malaysia, Jalan Sunsuria, Bandar Sunsuria, Sepang 43900, Selangor, Malaysia; 5Department of Veterinary Pathology and Microbiology, Faculty of Veterinary Medicine, Universiti Putra Malaysia, UPM Serdang 43400, Selangor, Malaysia; 6Department of Cell and Molecular Biology, Faculty of Biotechnology and Biomolecular Sciences, Universiti Putra Malaysia, UPM Serdang 43400, Selangor, Malaysia

**Keywords:** universal influenza A vaccine, *Macrobrachium rosenbergii* nodavirus, extracellular domain of matrix 2 protein, virus-like particle, H1N1, H3N2

## Abstract

Current seasonal influenza A virus (IAV) vaccines are strain-specific and require annual reconstitution to accommodate the viral mutations. Mismatches between the vaccines and circulating strains often lead to high morbidity. Hence, development of a universal influenza A vaccine targeting all IAV strains is urgently needed. In the present study, the protective efficacy and immune responses induced by the extracellular domain of Matrix 2 protein (M2e) displayed on the virus-like particles of *Macrobrachium rosenbergii* nodavirus (NvC-M2ex3) were investigated in BALB/c mice. NvC-M2ex3 was demonstrated to be highly immunogenic even in the absence of adjuvants. Higher anti-M2e antibody titers corresponded well with increased survival, reduced immunopathology, and morbidity of the infected BALB/c mice. The mice immunized with NvC-M2ex3 exhibited lower H1N1 and H3N2 virus replication in the respiratory tract and the vaccine activated the production of different antiviral cytokines when they were challenged with H1N1 and H3N2. Collectively, these results suggest that NvC-M2ex3 could be a potential universal influenza A vaccine.

## 1. Introduction

Influenza A virus (IAV) is one of the most menacing global killers in human history. The infamous Spanish Flu outbreak in 1918–1919 claimed at least 50 million lives worldwide and infected approximately one third of the world population. According to the World Health Organization (WHO), flu epidemics cause approximately 290,000 to 650,000 deaths annually, and the virus infects three to five million people globally. Despite advancements in the control of infectious diseases, the world is now more vulnerable to deadly pandemics than ever, partly due to the rapid expansion of global transport networks and unprecedented high rates of global travel frequency [1]. Furthermore, cross-species transmission of highly pathogenic avian IAV to humans raised concerns of avian influenza pandemics [2,3,4].

IAV is an enveloped virus with a genome composed of eight segmented, negative-sense RNAs encoding a total of 14 viral proteins [5]. Haemagglutinin (HA) and neuraminidase (NA) are the two most prominent surface glycoproteins of IAV, and are commonly used as the primary immunogenic components of current influenza vaccines [5,6]. During a natural infection, HA binds to the sialic acid receptors on the surface of host cells, and subsequently triggers membrane fusion [7,8]. On the other hand, NA is responsible for the release of virus progenies by cleaving the sialic acid from the infected cells [7,8]. Most antiviral medications against IAV infections, such as oseltamivir, zanamivir, and peramivir, target NA [9]. Nevertheless, exceptionally high mutation rates of HA and NA reduce the efficacy of these antiviral drugs. Thus, influenza A vaccines have to be reformulated annually to counter the rapidly mutating seasonal IAV [10]. The protective efficacy of current influenza A vaccines varies between 60% to 90% depending on the similarity between the immunogenic components recruited in an annual vaccine formulation and those of the circulating strains [5]. In the event of an unanticipated pandemic caused by genetic reassortments, seasonal influenza vaccines become less effective, in addition to the considerable amount of time and resources required for a hasty mass production of the pandemic vaccines [11]. Current prophylactic measures of IAV infections are impeded by rapidly mutating HA and NA. Therefore, employing a viral protein that is conserved across IAV strains is urgently needed for the development of a universal influenza A vaccine.

The matrix 2 (M2) protein of IAV is a homotetrameric proton channel responsible for the release of the viral RNAs, and budding of the virus progenies [12,13]. The M2 protein consists of 97 amino acid residues which can be divided into three segments: (i) an extracellular N-terminal segment, (ii) a transmembrane segment, and (iii) an intracellular C-terminal segment. The extracellular matrix 2 domain (M2e) is composed of 23 amino acid residues which are highly conserved across all human IAV strains, representing a potential candidate for the development of a universal influenza A vaccine [14]. Unlike HA and NA, the M2e is poorly immunogenic, and presents at a very low copy number on the surface of the virion. Even under natural IAV infection, or vaccination with the whole inactivated or live attenuated influenza A vaccine, no notable M2e-specific antibody titer could be detected [15]. In order to enhance the immunogenicity of M2e, various virus-like particles (VLPs) such as those of hepatitis B virus (HBV), human papilloma virus (HPV), papaya mosaic virus (PMV), tobacco mosaic virus (TMV), and T7 bacteriophage were genetically engineered to display M2e on the surface of these VLPs. Some of these chimeric VLPs were shown to induce protective immunity in mice against different IAV challenges [16,17,18,19,20,21]. Certain VLPs, such as those of RNA-phages Qβ and AP205, were shown to induce carrier-induced epitopic suppression (CIES) in mice, a condition that is characterized by the suppression of humoral immune responses directed against the target antigen conjugated to an immunogenic carrier due to pre-existing immunity against the carrier [22,23]. A similar concern emerged during the development of a human IAV vaccine using the HBV VLP as an antigen carrier [24]. Nevertheless, CIES in humans could be overcome by using VLPs derived from non-human infecting viruses such as the capsid of nodavirus [23].

VLPs produced by both plants and bacteria [18,19,20,21] had one copy of the M2e peptide fused to the viral capsid proteins. Conversely, the HBV VLPs [16,25], which had one to three copies of M2e peptides fused to the viral capsid protein, were demonstrated to have a better protective efficacy in proportion to the copy numbers of M2e peptide. We previously fused copies of M2e fragments to the C-terminal end of *Macrobrachium rosenbergii* nodavirus capsid protein (NvC), producing a chimeric protein (NvC-M2ex3) which self-assembles into VLPs (see Figure 2 in Yong et al. [26]). A subsequent immunization study demonstrated that NvC-M2ex3 induced strong M2e-specific antibody responses in the presence of the Freund’s adjuvant [26]. In this article, we report the protective efficacy and immune responses induced by NvC-M2ex3 without an adjuvant, in BALB/c mice against H1N1 (A/PR/8/34) and H3N2 (A/HK/8/68) challenges. H3N2 and H1N1 were used in this study because they are the most common IAV subtypes that emerge during annual flu seasons. Our results showed that the NvC-M2ex3 was highly immunogenic even in the absence of adjuvants. Higher anti-M2e antibody titers corresponded well with increased survival, reduced immunopathology and morbidity of the infected BALB/c mice. Following infection, the mice immunized with NvC-M2ex3 exhibited lower viral RNA copy numbers in the respiratory tract, suggesting limited viral replication. Moreover, different antiviral cytokines were activated in the mice immunized with NvC-M2ex3 and challenged with H1N1 and H3N2. Collectively, these results demonstrated that NvC-M2ex3 could be a potential universal influenza A vaccine.

## 2. Materials and Methods

### 2.1. Ethics Statement

All animal studies were approved and carried out in strict accordance with the guidelines of the Institutional Animal Care and Use Committee (IACUC, approval number R038/2015) of Universiti Putra Malaysia. All mice were housed in a temperature-controlled biosafety level 2 (BSL-2) animal facility with alternating 12 h of light and dark cycles.

### 2.2. Viruses

The virus strains, A/PR/8/34 (H1N1; ATCC^®^ VR-95™) and A/HK/8/68 (H3N2; ATCC^®^ VR-544™), were purchased from the American Type Culture Collection (ATCC). The mouse-adapted viruses were generated via serial lung passages in BALB/c mice. In brief, a virus stock (50 µL) containing H1N1 or H3N2 was inoculated intranasally into the nostril of lightly anaesthetized mice. Three days later, the mice were euthanized, and their lungs were harvested, homogenized, and clarified. Lung supernatants containing IAV were used for the following passage. H1N1 and H3N2 were adapted to BALB/c mice following a total of 8 and 20 passages, respectively. All procedures involving H1N1 or H3N2 were carried out in the BSL-2 laboratory.

### 2.3. Preparation of the Chimeric Protein and Immunization

Expression and purification of NvC and NvC-M2ex3 were carried out as previously described [26]. Protein concentrations were measured by the Bradford assay [27]. Groups of female BALB/c mice (n = 8) aged about 6 to 7 weeks were acclimatized for at least a week before being immunized subcutaneously with purified NvC-M2ex3 (0.34 mg/mL; 100 µL) 3 times at 3-week intervals. As a negative control, purified NvC (0.34 mg/mL; 100 µL) was also administered subcutaneously to the mice following the same immunization schedule. The blood of the mice was collected via submandibular bleeding before the first and second boosters and one week after the last booster. The collected blood samples were incubated at room temperature for 30 min, and the sera were collected after two consecutive centrifugations at 1000× *g* for 10 min at room temperature.

### 2.4. Immunogenicity of the Chimeric Protein

The anti-M2e antibody titers in serum samples were determined by enzyme-linked immunosorbent assay (ELISA). A synthetic peptide of M2e (amino acid sequence: KSLLTEVETPIRNEWGCRCNDSSD) was chemically synthesized (Genemed Synthesis Inc., San Antonio, TX, USA); dissolved in TBS buffer (50 mM Tris-HCl, 150 mM NaCl; pH 7.4), coated (2 µg/mL; 100 µL) on U-bottom 96-well microtiter plate wells and incubated overnight at 4 °C. The wells were rinsed once with TBST (TBS supplemented with 0.05% (*v*/*v*) Tween-20; pH 7.4) and blocked with milk diluent (200 µL; KPL, Gaithersburg, MD, USA) for 2 h at room temperature. After blocking, the wells were rinsed once with TBST. Serum samples (1:2000 dilution in TBS; 100 µL) were then added into the wells and incubated for 2 h with shaking followed by washing 3 times with TBST. Anti-mouse antibody conjugated to alkaline phosphatase (1:5000 dilution in TBS; KPL, Gaithersburg, MD, USA) was added into the wells and incubated for one hour. After washing the wells 3 times with TBST, p-Nitrophenyl phosphate (100 µL) was added, and further incubated for 20 min in the dark. Absorbance at wavelength 405 nm was measured using a VersaMax™ ELISA microplate reader (Molecular Devices, San Jose, CA, USA).

### 2.5. Challenges of Immunized Mice with Mouse-Adapted H1N1 and H3N2

At three weeks after the last injection, all mice immunized with NvC-M2ex3 or NvC were challenged intranasally with 4× median lethal dose (LD_50_) of mouse-adapted H1N1 or H3N2. LD_50_ [expressed in 50% egg infective dose (EID_50_)] of mouse-adapted H1N1 and H3N2 were 4.94 log_10_ EID_50_/mL ± 0.08 and 6.68 log_10_ EID_50_/mL ± 0.17, respectively. Survival rate, body weight loss and morbidity of the infected mice were monitored for 14 days. Mice with body weight loss more than 25% were euthanized by cervical dislocation according to IACUC guidelines. Morbidity of the mice was assessed based on the 4-point scale sickness score which evaluated ruffled fur (1 point), hunched back posture (1 point), and activity of mice (reduced: 1 point, severely reduced: 2 points) (see Appendix A). The sum of the score of each parameter is the total sickness score where the maximum score is 4 and 0 indicates healthy [28].

### 2.6. Determination of the Viral RNA Copy Numbers by Quantitative Polymerase Chain Reaction (qPCR)

At day-5 post-challenged with H1N1 or H3N2, oropharyngeal swabs and lungs of the mice were collected, homogenized, and clarified. The viral RNA in the lung supernatants and oropharyngeal swabs was extracted using the RNeasy Plus Mini Kit (Qiagen, Hilden, Germany) and reverse transcribed using the RevertAid H Minus Reverse Transcriptase (Thermo Fisher Scientific, Waltham, MA, USA) according to the manufacturer’s instructions. Specific primers (Forward primer: 5′-AAGACCAATCCTGTCACCTCTGA-3′; and Reverse primer: 5′-CAAAGCGTCTACGCTGCAGTCC-3′) which anneal to the *matrix* gene of IAV and the quantitative polymerase chain reaction (qPCR) conditions were adapted from Hopkins et al. [29]. The viral RNA copy number was determined by absolute quantification using a standard curve and calculated using Equation (1).
(1)$$\mathrm{Viral}\;\mathrm{RNA}\;\mathrm{copy}\;\mathrm{number}\;=\frac{\left(\mathrm{Amount}\;\mathrm{of}\;\mathrm{RNA}\;(g)\;*\;6.023\;\times 10^{23}\right)}{\left(330\times amplicon\;length\right)}$$
(6.023 × 10^23^ is Avogadro’s number)

### 2.7. Sandwich Enzyme-Linked Immunosorbent Assay (ELISA) for Quantification of Cytokines

Quantification of interferon-γ (IFN-γ), interleukin-12 (IL-12), and interleukin-6 (IL-6) in the lung supernatants of mice was performed using the DuoSet^®^ ELISA Development Systems following the manufacturer’s instructions. Anti-cytokine antibodies (100 µL) were first coated on a flat-bottom 96-well microtiter plate overnight at room temperature. After washing 3 times with washing buffer (137 mM NaCl, 2.6 mM KCl, 8 mM Na_2_HPO_4_, 1.5 mM KH_2_PO_4_, 0.05% (*v*/*v*) Tween-20; pH 7.4), the wells were blocked with 300 µL of PBS-bovine serum albumin (BSA) (137 mM NaCl, 2.6 mM KCl, 8 mM Na_2_HPO_4_, 1.5 mM KH_2_PO_4_, 1% (*w*/*v*) BSA; pH 7.4) for 2 h at room temperature. The wells were then washed 3 times with the same washing buffer followed by the addition of cytokine standards and the lung supernatants. Following 2 h of incubation at room temperature, unbound cytokine standards or lung supernatant were removed by washing the wells with washing buffer. Antibodies for cytokine detection were then added to the wells and incubated for 2 h at room temperature. Subsequently, the wells were washed again as described above, and the streptavidin conjugated horseradish peroxidase (HRP) enzyme (100 µL) was added to the wells followed by incubation at room temperature for 20 min. Following another washing step, substrate solution (1:1 mixture of H_2_O_2_ and tetramethylbenzidine; 100 µL) was added to the wells and incubated for 20 min at room temperature. Lastly, sulphuric acid (1 M; 50 µL) was added to the wells to terminate the reaction and the absorbance at wavelength 450 nm was measured with the VersaMax^TM^ ELISA Microplate Reader (Molecular Devices, San Jose, CA, USA).

### 2.8. Hematoxylin and Eosin Staining

The lungs of mice were excised and fixed in 10% (*v*/*v*) formalin and embedded in parafilm wax. Fixed tissues were sectioned (50 µm thick) and stained with hematoxylin and eosin (H and E). Parameters used in scoring the total lung inflammation were bronchiolitis, edema, and interstitial inflammation. The severity of each parameter was scored in a blinded manner on a scale of 0 to 4, in which 0 indicates absent, and 1 to 4 indicate increasing severity. The sum of the score of each parameter is the total lung inflammation score and the maximum score is 12 [30].

### 2.9. Statistical Analysis

The differences in immunogenicity, weight loss, and morbidity scores among animal groups were analyzed using one-way analysis of variance (ANOVA) with Duncan’s multiple range test. Variations between different treatment groups with respect to viral RNA copy numbers, cytokine levels, and lung inflammatory scores were analyzed using the independent Student’s *t*-test. The Kaplan–Meier method was used for the survival analysis using the log-rank test. A *p*-value of lower than 0.05 is considered significant, lower than 0.01 is considered very significant, lower than 0.001 is considered highly significant, and lower than 0.0001 is considered extremely significant.

## 3. Results

### 3.1. Immunogenicity of the Chimeric Protein in BALB/c Mice

BALB/c mice were immunized subcutaneously with NvC or NvC-M2ex3. Serum samples of the mice were collected three weeks after the primary administration, three weeks after the first booster, and one week after the second booster. Figure 1 shows the anti-M2e antibody titer in the immunized mice. After the primary injection, no apparent difference was observed in the anti-M2e antibody titer between the mice immunized with NvC-M2ex3 and the control group injected with NvC. Nevertheless, a distinct spike in the anti-M2e antibody titer was detected in the group immunized with NvC-M2ex3 three weeks after the first booster, and a further increase in the antibody level was observed upon administration of the second booster. Conversely, the control groups immunized with NvC lacking the M2e epitopes did not show any significant increase in the anti-M2e antibody titer. In general, the result reveals that NvC-M2ex3 is highly immunogenic in BALB/c mice, and the anti-M2e antibody titer increased with the number of injections.

### 3.2. Survival and Morbidity of Mice Challenged with Influenza A Viruses

Three weeks after the last booster, the mice (n = 8) were infected intranasally with 4x LD_50_ of mouse-adapted H1N1 or H3N2. The infected mice were monitored for survival, weight loss, and morbidity up to 14 days post-infection. All mice immunized with NvC-M2ex3 survived the lethal challenges with H1N1 and H3N2, whereas mice treated with NvC succumbed to the infections and died at day six or seven post-infection (Figure 2A,B). The mice immunized with NvC-M2ex3 showed less body weight loss and lower sickness scores when challenged with lethal mouse-adapted H1N1 (Figure 2C,E). Interestingly, after exposure to lethal mouse-adapted H3N2, the mice vaccinated with NvC-M2ex3 did not show significant body weight loss or morbidity (Figure 2D,F). The onset of symptoms of H1N1 and H3N2 infections began at day three and four respectively, and peaked at day seven post-infections. The mice immunized with NvC-M2ex3 started to convalesce from the infection at day nine and resolved completely at day-12 post-challenged with H1N1. On the other hand, the mice immunized with NvC-M2ex3 and challenged with H3N2 fully recovered a day after the peak of infection (day eight post-infection) (Figure 2E,F). All mice immunized with NvC exhibited significant weight loss, severe morbidity, and inevitable death following the infections with lethal H1N1 or H3N2. These results demonstrate the protective efficacy of NvC-M2ex3 against IAV H1N1 and H3N2 infections.

### 3.3. Viral RNA Copy Number in the Lungs and Oropharynges of the Infected Mice

Viral RNA copy number in the lungs and oropharynges of the infected mice was determined by qPCR using primers specifically targeting the *matrix* gene of IAV. The viral RNA copy number was calculated using Equation (1). The mice immunized with NvC-M2ex3 exhibited a 10-fold reduction of viral RNA copy number in the lungs compared to the control group after being challenged with H1N1 or H3N2 (Figure 3A). This result demonstrates the infection-permissive nature of NvC-M2ex3 which allows limited virus replication instead of total inhibition. The oropharynges of the mice immunized with NvC-M2ex3 also showed a 10-fold reduction in the viral RNA copy number compared to the control group immunized with NvC (Figure 3B). The presence of IAV in the oropharynges of the mice indicates viral shedding and potential transmission of the virus between animals via direct close contact.

### 3.4. Cytokines Concentrations in the Lungs of the Mice

The concentrations of interferon-γ (IFN-γ), interleukin-12 (IL-12), and interleukin-6 (IL-6) in the lungs of infected mice were quantified using ELISA. After H1N1 infection, significantly higher levels of IFN-γ and IL-12 were observed in the mice immunized with NvC-M2ex3 as compared to the control group, but no obvious difference was observed in the level of IL-6 (Figure 4A). Intriguingly, when NvC-M2ex3-immunized mice were challenged with H3N2, the concentrations of IFN-γ and IL-6 were significantly lower than those mice immunized with NvC, despite a higher level of IL-12 being observed (Figure 4B). These results show that the mice infected by different strains of IAV had different cytokine responses.

### 3.5. Histopathological Analysis

At day five post-infection with H1N1 or H3N2, the lungs of the mice were harvested, fixed, sectioned, and stained with H and E. The degree of lung inflammation was scored based on the severity of bronchiolitis, edema, and interstitial inflammation. Infections by different strains of IAV resulted in varying magnitudes of lung inflammation. The mice immunized with NvC-M2ex3 experienced mild bronchiolitis, edema, and interstitial inflammation when challenged with H1N1 (Figure 5(Ai)) but demonstrated a very mild bronchiolitis when they were challenged with H3N2 (Figure 5(Bi)), suggesting a remarkable resistance to H3N2 infection. Both control groups exhibited severe bronchiolitis, moderate edema, and interstitial inflammation (Figure 5(Aii,Bii)). The mice immunized with NvC-M2ex3 also scored lower in total lung inflammation relative to those immunized with NvC after being infected by H1N1 or H3N2 (Figure 6). Histopathological scores of the lungs of mice are in good agreement with the survival, morbidity, and body weight loss of the mice.

## 4. Discussion

VLPs are multimeric nanostructures composed of viral structural proteins with inherent self-assembly properties. Morphologically, VLPs resemble the authentic viruses but are devoid of viral genetic materials, rendering them non-replicative and non-infectious. Attributed to their intrinsic adjuvanticity, several VLPs were reported to be capable of inducing strong B-cell and T-cell responses even in the absence of adjuvants [31]. In an attempt to produce a universal influenza A vaccine, we previously constructed chimeric VLPs derived from NvC that were genetically fused with three copies of the M2e epitope at its C-terminal region [26]. Although NvC-M2ex3 was demonstrated to be immunogenic, its protective efficacy has not been investigated in vivo. Therefore, in the present study, we investigated the protective efficacy of NvC-M2ex3 against H1N1 and H3N2 infections. BALB/c mice were immunized subcutaneously with NvC-M2ex3 three times at three-week intervals. After the primary injection, the anti-M2e antibody response was insignificant and indistinguishable from the control group. However, a spike of anti-M2e antibody titer was detected three weeks after a booster was administered and a further increase was observed one week after the second booster. In our previous study, we administered NvC-M2ex3 into BALB/c mice in the presence of a complete Freund’s adjuvant using the same dose and same immunization schedule as in this study. Our previous results demonstrated a significant increase in anti-M2e antibody titer just three weeks after the primary injection and a maximum titer was achieved after the first booster, indicating that the maximum antibody response could be attained with a lower injection number in the presence of adjuvants [26]. Although a complete protective mechanism of M2e-based vaccine remains poorly understood, passive transfer of the anti-M2e antibody was demonstrated to be able to protect the recipients against viral infection, suggesting the importance of this antibody [32]. Furthermore, the anti-M2e antibody was also reported to confer protection against influenza virus infection via antibody-dependent cytotoxicity and antibody-mediated phagocytosis in Fcγ receptor knockout mice [33,34]. In the present study, a higher titer of anti-M2e antibody corresponded well with increased survival, reduced immunopathology, and morbidity of IAV-infected mice.

Three weeks after the last booster, all mice were challenged with lethal doses of mouse-adapted H1N1 or H3N2. Remarkably, all mice immunized with NvC-M2ex3 were 100% protected against both H1N1 and H3N2 challenges compared to the control groups with 100% mortality. Intriguingly, the mice immunized with NvC-M2ex3 were shown to be more resistant to H3N2 than H1N1 infection, suggesting the protective mechanisms induced by NvC-M2ex3 were more effective against H3N2 than H1N1. Variation in the mechanism of pathogenicity of different strains of IAV may have contributed to different severity in the mice immunized with NvC-M2ex3. Previous studies indicated that it is not uncommon for M2e-based vaccines to have different protective efficacies against different strains of IAV [16,35,36]. Previously, HBV, PMV, and T7 bacteriophage VLPs displaying M2e epitope were also demonstrated to protect immunized mice against lethal IAV infections but generally required additional adjuvants to achieve 100% protection [19,21,25]. No apparent weight loss or physical sign of illness was observed in mice vaccinated with NvC-M2ex3 after H3N2 challenges despite slight histological changes being observed in the lungs. Nevertheless, a notable amount of viral RNA was detected in the lungs of all mice challenged with H1N1 or H3N2 with a viral RNA copy number approximately 10-fold lower in the group immunized with NvC-M2ex3 compared to the control groups. As this was a relatively conservative reduction of viral RNA copy number, it is believed that the immune responses triggered by NvC-M2ex3 suppress but do not inhibit virus replication. This result suggests the infection-permissive nature of NvC-M2ex3 which allows controlled virus replication, which is in good agreement with Schotsaert et al. [35], who demonstrated similar limited virus replication in the lungs of the mice vaccinated with an M2e-based vaccine. The infection-permissive nature of an M2e-based vaccine is believed to allow natural infection of the host by IAV with reduced morbidity and induction of a more effective T-cell response directed against conserved internal viral proteins. Each new infection exposes the host to new viral antigens, which enables natural update of the host immune responses against different epitopes, inducing a natural heterosubtypic protection in the long run [35].

Influenza virus is transmitted primarily via close contact, aerosols, or droplets [37]. A hospitalized patient with severe IAV infection can shed a detectable and infectious amount of virus for a longer period of time [38]. In general, detection of influenza viral RNA in the respiratory airway indicates the viral infection, and a high viral load in the upper respiratory airway is associated with an increased risk of transmission, which often correlates with the severity of the disease [39]. In the present study, the mice immunized with NvC-M2ex3 experienced mitigated morbidity and a reduced viral RNA copy number in their oropharynges compared to the control groups after being challenged with H1N1 or H3N2, suggesting the suppression of viral shedding.

The mice immunized with NvC-M2ex3 demonstrated distinct cytokine profiles when they were challenged with different strains of IAV. Upon infection with H1N1, the mice immunized with NvC-M2ex3 exhibited a higher level of IFN-γ and IL-12 in the lungs compared to the control groups. Positive correlation between IL-12 and IFN-γ in influenza infection was also reported by Monteiro et al. [40]. IL-12 enhanced IFN-γ secretion which reduced pathogenesis and replication of IAV at the early stages of the viral infection [41,42]. Intriguingly, when the mice that had been vaccinated with NvC-M2ex3 were challenged with H3N2, a lower concentration of IFN-γ was observed compared to the control groups, despite a higher level of IL-12 also being detected, suggesting dysregulation of IFN-γ. The balance of inhibitory and stimulatory signals directed to IFN-γ-producing cells along with IL-12 could be the underlying cause that leads to this outcome. Similar cytokine profiles were observed in the peripheral blood mononuclear cells of patients infected with multidrug-resistant tuberculosis and cord blood-derived mononuclear cells of neonates stimulated with *Staphylococcus aureus* [43,44]. Although IL-6 was demonstrated to correspond with the lung pathogenesis and disease severity induced by IAV in mice, it was also reported to promote neutrophil survival, virus clearance, and mitigated lung inflammation [45,46,47]. In the present study, the mice immunized with NvC-M2ex3 exhibited a lower level of IL-6 in the lungs compared to the control group when challenged with H3N2, but no significant difference was observed after H1N1 infection. On the other hand, histopathological analysis of the lungs of infected mice also suggests a reduced immunopathology in the mice immunized with NvC-M2ex3 after IAV challenges, but did not correspond with the mentioned pro-inflammatory cytokine levels in the lungs, indicating that other cytokines might have played dominant roles in mediating the immunopathology. Secretion of pro-inflammatory cytokines potentially confers early protection against IAV infections, but excessive activation could lead to massive tissue damage.

In comparison to conventional egg-based vaccine production, the approach used in this study is faster, less laborious, and more cost-effective. Apart from the universal antibody-based protective immunity elicited by NvC-M2ex3, its infection-permissive nature can potentially induce the development of natural heterosubtypic protection, contrary to the current licensed inactivated influenza vaccine which was reported to hamper the cross-protectivity of T-cell immunity upon subsequent infection [35,48]. In addition, NvC-M2ex3 was engineered from *M. rosenbergii* nodavirus, a virus that only infects the giant freshwater prawn, thereby eliminating the risk of CIES in humans [22,49]. Although suppressed viral shedding was detected in the oropharynges of the mice immunized with NvC-M2ex3, this may not reflect the risk of virus transmission in mice because influenza A virus does not spread effectively in mice and was reported to be highly dependent on the virus strains and mouse models [50].

## 5. Conclusions

In summary, this study demonstrated a 100% protective efficacy of NvC-M2ex3 against lethal H1N1 and H3N2 infections in BALB/c mice by eliciting M2e specific humoral immune responses while suppressing the replication of IAV. Activation of antiviral cytokines could also potentially contribute to early protection against influenza infections. Recommendations for future studies include investigation of the mechanisms of protection, passive serum transfer, long-term protection, and challenges with different strains of influenza A viruses.

## 6. Patent

A patent entitled “An Influenza A Vaccine” (application no: PI 2018702779) was filed on 8 August 2018.

## Figures and Tables

**Figure 1 vaccines-07-00091-f001:**
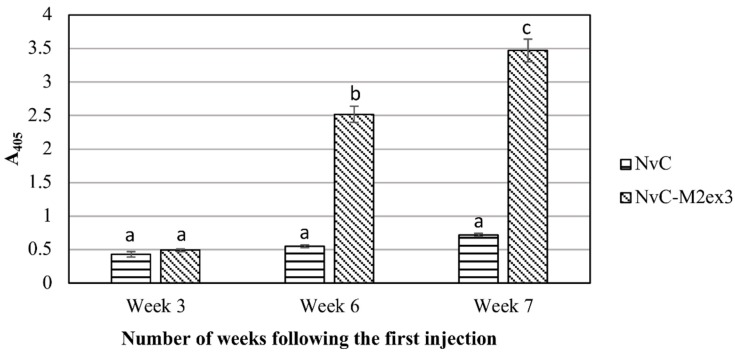
Immunogenicity of the chimeric protein in BALB/c mice. Serum samples of BALB/c mice immunized with NvC or NvC-M2ex3 were collected at three weeks after the primary injection, three weeks after the first booster, and one week after the second booster by submandibular bleeding. Serum samples (1:2000 dilution) from mice (n = 8) of respective groups were pooled and analyzed by enzyme-linked immunosorbent assay (ELISA). The microtiter plate wells were coated with the M2e synthetic peptide and the anti-M2e antibody reacted with the peptide was detected with the anti-mouse antibody conjugated to alkaline phosphatase. a, b, and c indicate statistical significance (*p* < 0.001) of the results within each time point and when compared to other time points.

**Figure 2 vaccines-07-00091-f002:**
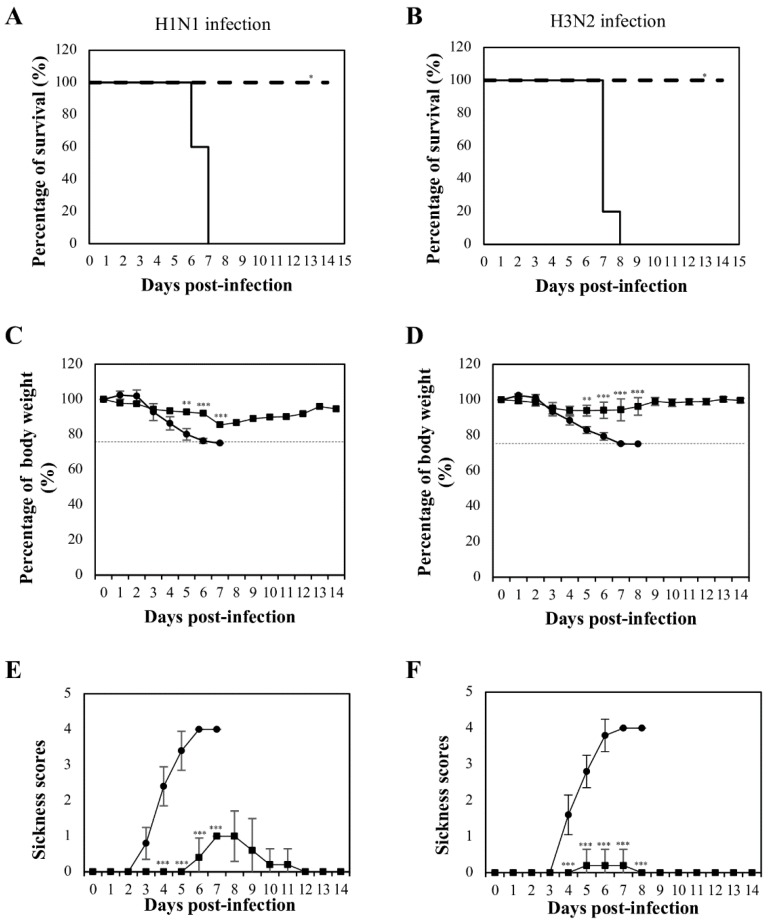
Mortality and morbidity of BALB/c mice after lethal influenza A virus infections. Survival rate of mice infected with (**A**) H1N1 and (**B**) H3N2. Solid and dashed lines indicate the survival rates of the mice immunized with NvC and NvC-M2ex3, respectively. Percentage of body weight of mice infected with (**C**) H1N1 and (**D**) H3N2. Dashed lines indicate endpoint of the animals. Morbidity of mice infected with (**E**) H1N1 and (**F**) H3N2. A complete protection against H1N1 or H3N2 infection was observed in the mice immunized with NvC-M2ex3. After infection with mouse-adapted H3N2, the mice vaccinated with NvC-M2ex3 did not demonstrate significant weight loss and morbidity. On the contrary, all mice immunized with NvC succumbed to the infections. (●) and (■) indicate the percentage of body weight and morbidity of the mice immunized with NvC and NvC-M2ex3, respectively. The asterisks (* *p* < 0.01, ** *p* < 0.001, *** *p* < 0.0001) indicate statistical significance of the results between different treatment groups.

**Figure 3 vaccines-07-00091-f003:**
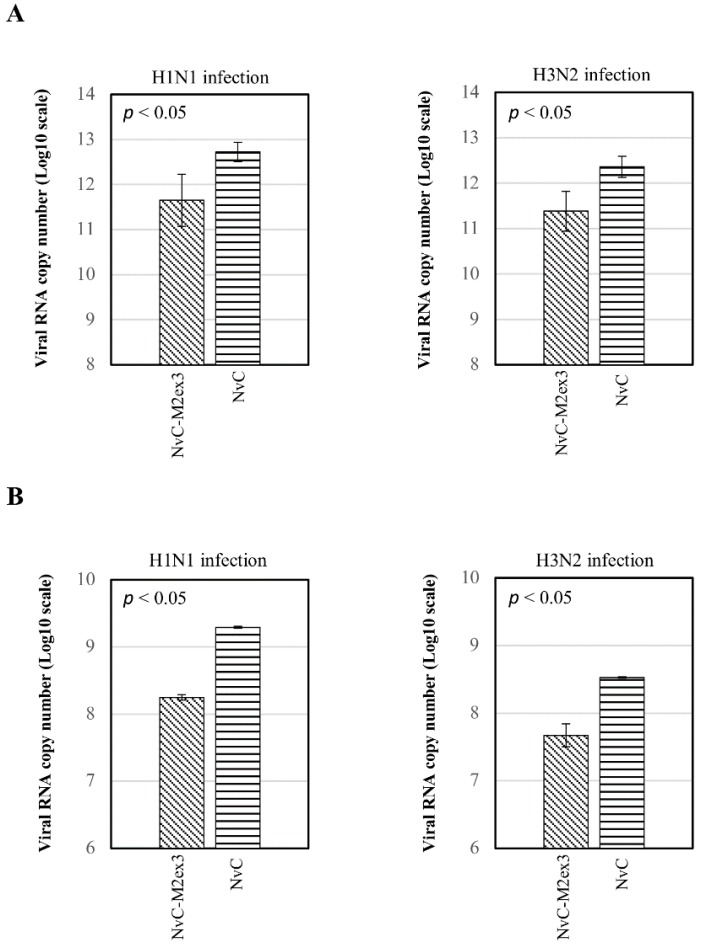
Viral RNA copy numbers in the lungs and oropharynges of infected mice. At day five post-infection, the lungs and oropharyngeal swabs of the infected mice (n = 3) were obtained. The viral RNA was then extracted, reverse transcribed, and quantified using quantitative polymerase chain reaction (qPCR). (**A**) Higher viral RNA copy numbers (10-fold) were detected in the lungs of the mice immunized with NvC relative to those immunized with NvC-M2ex3 after being challenged with H1N1 or H3N2. These results demonstrate the potential of NvC-M2ex3 in limiting virus replication. (**B**) Detection of the viral RNA of H1N1 or H3N2 in the oropharynges of the mice suggests viral shedding and possible transmission between animals. All the NvC-M2ex3-immunized mice exhibited a reduced viral shedding in both lethal H1N1 and H3N2 infections. The differences between treatment groups are significant with *p* < 0.05.

**Figure 4 vaccines-07-00091-f004:**
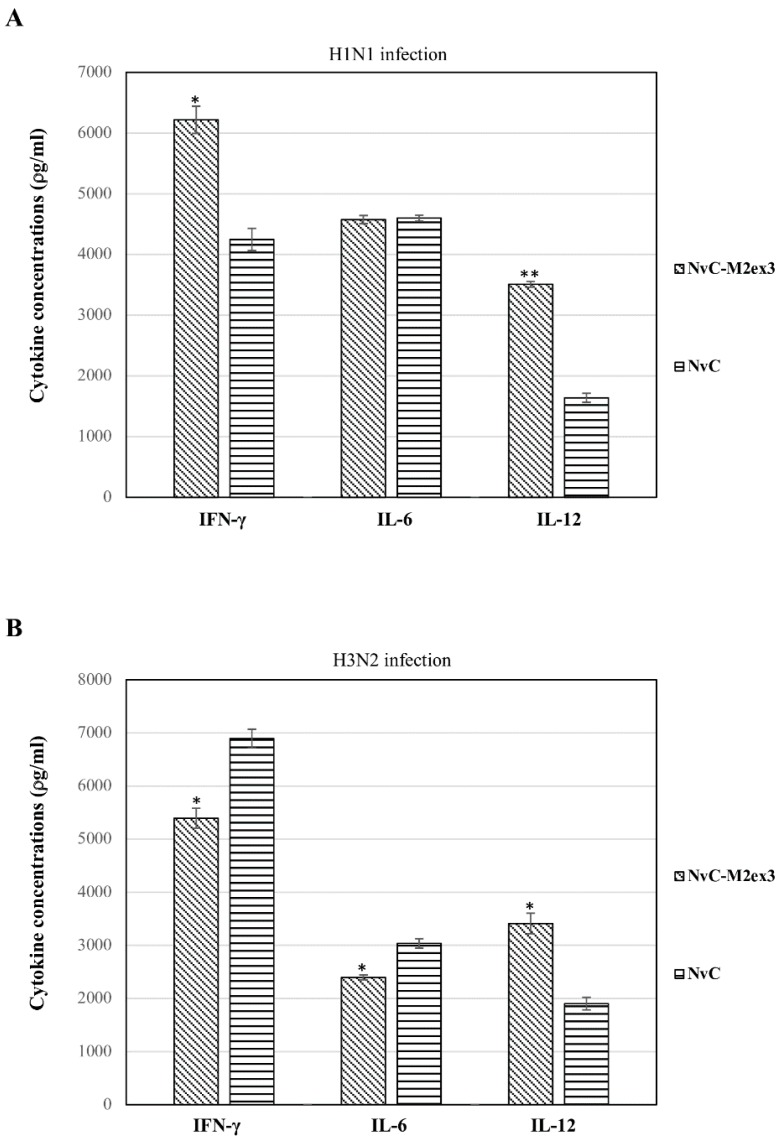
Concentration of cytokines in the lungs of mice (n = 3) infected by H1N1 and H3N2. The mice vaccinated with NvC-M2ex3 demonstrated different cytokine profiles after being challenged with H1N1 and H3N2. (**A**) A higher level of IFN-γ and IL-12 was detected in the lungs of the mice immunized with NvC-M2ex3 compared to the control group after being challenged with H1N1. (**B**) On the contrary, when the mice were challenged with H3N2, they exhibited lower concentrations of IFN-γ and IL-6, although a higher level of IL-12 was observed. The asterisks (* *p* < 0.05, ** *p* < 0.01,) indicate statistical significance of the results between different treatment groups.

**Figure 5 vaccines-07-00091-f005:**
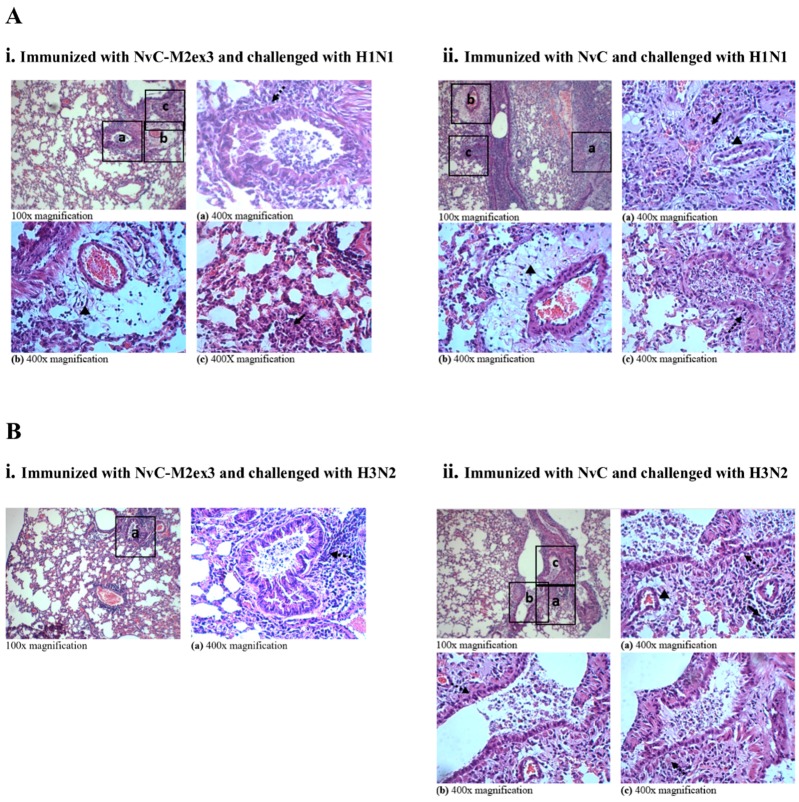
Hematoxylin and Eosin (H and E) staining of the lungs of mice (n = 3) immunized with the chimeric proteins and challenged with influenza A virus H1N1 or H3N2. (**A**) The lungs of mice immunized with (i) NvC-M2ex3 or (ii) NvC at day five post-infection with H1N1. (**B**) The lungs of mice immunized with (i) NvC-M2ex3 or (ii) NvC at day five post-infection with H3N2. Insets with letters (a, b, c) are projected at 400x magnification as (a), (b), and (c). Dashed and solid arrows indicate bronchiolitis and interstitial inflammation respectively, and triangles indicate edema, n = 3.

**Figure 6 vaccines-07-00091-f006:**
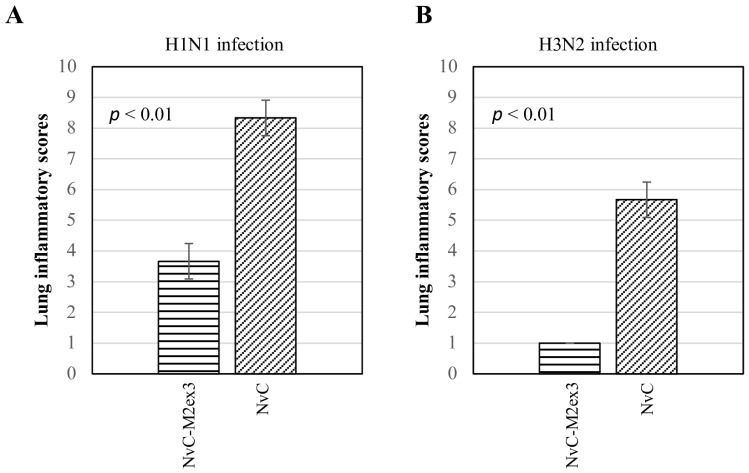
Lung inflammatory scores of the mice (n = 3) challenged with influenza A virus. A lower lung inflammatory score was observed in the mice immunized with NvC-M2ex3 after being challenged with (**A**) H1N1 or (**B**) H3N2. All mice immunized with NvC experienced a higher lung inflammatory score which corresponded well with increased morbidity and reduced survival. The differences between the treatment groups are significant with *p* < 0.01, n = 3.

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
