# Peer review of "An Influenza A Vaccine Based on the Extracellular Domain of Matrix 2 Protein Protects BALB/C Mice Against H1N1 and H3N2"

_vaccines, 2019, doi:10.3390/vaccines7030091_

Round 1

Reviewer 1 Report

The authors present a clear and important work on the immunogenicity and protective capacity of a nanoparticles displaying the conserved ectodomain of the M2 protein of influenza.

The work is well written and important for influenza vaccine development in terms of producing more effective influenza vaccines.

I have only minor points:

Line 20. "by extracellular" should be "by the extracellular"

Line 49-50. Should "these two glycoproteins" be "NA".  The drugs target the NA.

The study is well written and designed and appropriate for publication.

Author Response

Comments and Suggestions for Authors

The authors present a clear and important work on the immunogenicity and protective capacity of a nanoparticles displaying the conserved ectodomain of the M2 protein of influenza.

The work is well written and important for influenza vaccine development in terms of producing more effective influenza vaccines.

I have only minor points:

1.    Line 20. "by extracellular" should be "by the extracellular"

Response: "by extracellular" has been changed to "by the extracellular". (page 1, line 24)

2.    Line 49-50. Should "these two glycoproteins" be "NA".  The drugs target the NA.

Response: Many thanks for pointing out this mistake. We have now changed "these two glycoproteins" to "the NA".  (page 2, line 53)

3.    The study is well written and designed and appropriate for publication.

Response: Many thanks for your constructive comments.

Reviewer 2 Report

In this manuscript, Ong et al., tested the protectiveness of a M2e-based vaccine that used self-assembly nodavirus-like particles displaying the M2e tandem repeats in mice. The vaccine platform was previously developed by the group, and the authors showed that three immunizations of the vaccine without adjuvant elicited protective antibody responses against a H1N1 strain and a H3N2 strain in mice. It was also shown that the vaccine reduced lung viral load and immunopathology but increased antiviral cytokines levels such as IL-12. The nodavirus like particle platform showed novelty in M2e vaccine design in their previous study, however the main criticism is that it wasn’t clear what are the advantages of nodavirus like particle M2 vaccine over the existing M2e-VLPs vaccines.

Comments:

1. What are the advantage of nodavirus like particle over other virus like particle M2e vaccine platforms? For example, does it include higher copy number of M2e? Does it induce higher level of antibody response? Does it induce longer-lasting antibody responses? Are there any evidence of negative effects of preexisting immunity to other virus-like particles?

2. It might be helpful to include a figure that illustrate how the NvC-M2ex3 is constructed again

3. Have the authors done a challenge study using a virus that does not have a completely matched M2e domain such as pH1N1 (A/Cal09) to show cross-protection?

4. Have the authors measured lung titers by infectious particles? Because defective interfering particles might still have some intact segments of vRNA.

5. It’s better to make table 1 into a figure

6. It’ll be easier for the readers if including a subtitle (immunized or mock with challenge virus) on top of each panel (I, ii, iii, iv) of histopathological analysis (Figure 4)

7. Line 290, did the authors do correlation analysis to draw this conclusion?

Author Response

Comments:

1.    What are the advantage of nodavirus like particle over other virus like particle M2e vaccine platforms? For example, does it include higher copy number of M2e? Does it induce higher level of antibody response? Does it induce longer-lasting antibody responses? Are there any evidence of negative effects of preexisting immunity to other virus-like particles?

Response: Many thanks for your comments. The antibody levels presented in this study are relative values to the controls. As a result, we cannot compare the triggered antibody levels in this study with those using different VLP platforms unless the experiments were carried out using the same experimental condition. The long term protection provided by NvC-M2ex3 has not been investigated, therefore we cannot draw a conclusion at this stage.

The construct used in this study comprises 3 copies of M2e similar to a previous study using the HBV VLP as a platform (De Filette et al. 2005). Although the HBV capsid-based M2e has been shown to protect the immunised mice against lethal virus challenge, the absence of carrier induced epitopic suppression (CIES) due to the pre-existing immunity against HBV was not reported. CIES is not a concern in this study because MrNV is not a virus that infects humans. We have included the benefit of using MrNV capsid derived VLP as IAV vaccine development in the Introduction (page 2, lines 74 to 86) and Results sections (page 13, lines 492-493).

2.    It might be helpful to include a figure that illustrate how the NvC-M2ex3 is constructed again

Response: Many thanks for your suggestion. Yong et al (2015) described clearly how the NvC-M2ex3 was constructed. In order make it clearer, we have now mentioned that the illustration of the construct can be found in Figure 2; Yong et al (2015). (page 2, line 88)

3.    Have the authors done a challenge study using a virus that does not have a completely matched M2e domain such as pH1N1 (A/Cal09) to show cross-protection?

Response: No, we did not challenge the mice with pH1N1 (A/Cal09) due to inaccessibility to this pandemic strain. Furthermore, we also do not have  access to a higher level of biosafety containment facility (biosafety level 3).

4.    Have the authors measured lung titers by infectious particles? Because defective interfering particles might still have some intact segments of vRNA.

Response: We have not measured the lung titers by infectious particles because the virus strains that we used for animal challenges are not adapted to cell-line.

5.    It’s better to make table 1 into a figure.

Response: This is indeed a good suggestion. We have now converted Table 1 into a figure (Figure 4). (page 9, lines 307-346)

6.    It’ll be easier for the readers if including a subtitle (immunized or mock with challenge virus) on top of each panel (I, ii, iii, iv) of histopathological analysis (Figure 4).

Response: We have now included the subtitle on top of each panel of histopathological analysis. Figure 4 becomes Figure 5 in the revised manuscript (page 10, lines 359-398).

7.    Line 290, did the authors do correlation analysis to draw this conclusion?

Response: No, correlation analysis was not performed. To avoid confusion, we have now changed the word “correlate” to “in good agreement”. (page 10, line 358)

8.    Moderate English changes required.

Response: The English has been edited by a native English speaker.  

Reviewer 3 Report

The limited breadth of protection offered by current seasonal influenza virus vaccines has led many to study antigens capable of eliciting more broad protection. While M2e is normally poorly immunogenic, immunogenicity has previously been reported to improve when presented in the context of VLPs. Here, the authors describe an M2e vaccine based on Macrobrachium rosenbergii nodavirus VLPs. The authors show that this vaccine is capable of protecting mice from lethal challenge with H1N1 and H3N2 viruses. Overall, the manuscript is straighforward and the science is mostly sound. Novelty, however, is quite limited as many groups have previously used similar approaches to elicit M2e antibodies. The authors should attempt to provide more mechanistic data to explain the observed protection - since they have already published a paper on immunogenicity of this vaccine. The title should also be edited to more accurately reflect the authors results. Specific recommendations can be found below.

Major Concerns:

1. The title of this article is clearly sensational and must be changed. Challenge of mice with 2 viruses is clearly insufficient to claim that this vaccine may be "universal." This reviewer has no problem if the authors would like to speculate as to the breadth of the vaccine in the body of the manuscript - but the title should be limited to reflect the actual finding presented in the manuscript: "An Influenza A Virus Vaccine Based on the Extracellular Domain of Matrix 2 Protein Protected BALB/C Mice Against H1N1 And H3N2."

2. The overall advance in knowledge that this study adds is extremely limited, since the mechanism of protection is not explored in any satisfying detail and the authors have previously published immunogenicity analyses. The authors seem to focus on antibody-mediated responses in their measurements of immunogenicity in the current manuscript. Thus, it would be of great interest to perform a serum transfer study to determine whether antibodies are sufficient to mediate protection.

3. Figure 3: Measurement of RNA copy number is not a valid measure of "viral load" - which should be assessed by plaque assay (or another measure of replication-competent virus) of lung homogenates. It would be preferable for the authors to repeat this experiment and measure actual viral titers in the lungs. Alternatively (but less desirably), the authors could re-write the corresponding sections and replace "viral load" with "RNA copy number" for accuracy.

Minor Concerns:

1. It would be more accurate to use the word "reformulated" instead of "reconstituted" when referring to the annual inclusion of new strains in influenza virus vaccines.

2. Lines 112-113: Please use (x g) when describing centrifugation conditions (instead of rpm, which varies based on rotor size).

3. Lines 179-181: Did the authors intend to state thatp-values "0.01" (not 0.001) were considered very significant, and 0.001 were considered extremely significant (not 0.0001)? Please clarify whether this was a typographical error?

4. Figure 1 legend: Please note serum dilution used for ELISA in figure legend for clarity.

5. Figure 2: It would be helpful for readers if the challenge strains were noted on each panel (as in Figure 3).

6. Figure 2 C & D: It would be helpful to include a dashed line to indicate the percentage weight loss at which endpoint was reached.

7. Table 1: This data is inconsistent and adds little value, in the opinion of this reviewer, to the manuscript. Recommend omitting. If the authors insist on keeping this table, the number of samples analyzed should be indicated.

8. Please comment on whether pathological analysis was performed in a blinded manner?

9. Figure 4Bi. It would be preferable, for consistency, if the authors chose a section in which 3 inlays were selected, as with all other panels in the figure.

10. Figure 4&5 legends: How many animals is this data representative of?

Author Response

Major Concerns:

1.    The title of this article is clearly sensational and must be changed. Challenge of mice with 2 viruses is clearly insufficient to claim that this vaccine may be "universal." This reviewer has no problem if the authors would like to speculate as to the breadth of the vaccine in the body of the manuscript - but the title should be limited to reflect the actual finding presented in the manuscript: "An Influenza A Virus Vaccine Based on the Extracellular Domain of Matrix 2 Protein Protected BALB/C Mice Against H1N1 And H3N2."

Response: Many thanks for your kind suggestion. We have now changed the title to “"An Influenza A Vaccine Based on the Extracellular Domain of Matrix 2 Protein Protects BALB/c Mice Against H1N1 and H3N2". (page 1, lines 2-4)

2.    The overall advance in knowledge that this study adds is extremely limited, since the mechanism of protection is not explored in any satisfying detail and the authors have previously published immunogenicity analyses. The authors seem to focus on antibody-mediated responses in their measurements of immunogenicity in the current manuscript. Thus, it would be of great interest to perform a serum transfer study to determine whether antibodies are sufficient to mediate protection.

Response: This is indeed a good suggestion. However, we are unable to perform this experiment due to limited funding and also expiry of ethical approval. A new application will take approximately 2 months to get the approval. We hope that acceptance of this paper will provide knowledge of antibody-mediated response to other researchers/readers with similar interest. Nevertheless, we have added this suggestion as a future study in the conclusion section. (page 13, line 503)

3.    Figure 3: Measurement of RNA copy number is not a valid measure of "viral load" - which should be assessed by plaque assay (or another measure of replication-competent virus) of lung homogenates. It would be preferable for the authors to repeat this experiment and measure actual viral titers in the lungs. Alternatively (but less desirably), the authors could re-write the corresponding sections and replace "viral load" with "RNA copy number" for accuracy.

Response: In addition to insufficient funding and expiry of ethical approval, we also could not measure the viral load by plaque assay because the virus strains used for the animal challenges were not adapted to an animal cell-line. Therefore, we have now replaced the “viral load” with “RNA copy number” for a better accuracy as suggested by the reviewer (page 8, lines 287 and 290) and also within the text (Page 3, line 96; page 4, line 149 and 157; page 5, line 190; page 12, lines 445, 447, and 462).   

4.    Comment: Extensive editing of English language and style required

Response: The English has been edited by a native English speaker.  

Minor Concerns:

1.    It would be more accurate to use the word "reformulated" instead of "reconstituted" when referring to the annual inclusion of new strains in influenza virus vaccines.

Response: We have now replaced "reconstituted" with “reformulated” in the revised manuscript. (page 2, line 54) 

2.    Lines 112-113: Please use (x g) when describing centrifugation conditions (instead of rpm, which varies based on rotor size).

Response: We have now converted “rpm” to “xg”. (page 3, line 125)

3.    Lines 179-181: Did the authors intend to state that p-values "0.01" (not 0.001) were considered very significant, and 0.001 were considered extremely significant (not 0.0001)? Please clarify whether this was a typographical error?

Response: This is not a typographical error. To clarify this, we have now rephrased the sentence to “A p-value of lower than 0.05 is considered significant, lower than 0.01 is considered very significant, less than 0.001 is considered highly significant, and lower than 0.0001 is considered extremely significant”. (page 5, lines 192-194)

4.    Figure 1 legend: Please note serum dilution used for ELISA in figure legend for clarity.

Response: We have now added the serum dilution used for ELISA in the figure legend (page 5, line 212)

5.    Figure 2: It would be helpful for readers if the challenge strains were noted on each panel (as in Figure 3).

Response: We have now indicated the challenge strains in each panel. (page 8)

6.    Figure 2 C & D: It would be helpful to include a dashed line to indicate the percentage weight loss at which endpoint was reached.

Response: Dashed lines have been added to indicate the endpoints. (please see revised Figure 2). 

7.    Table 1: This data is inconsistent and adds little value, in the opinion of this reviewer, to the manuscript. Recommend omitting. If the authors insist on keeping this table, the number of samples analyzed should be indicated.

Response: As suggested another reviewer, we have now converted Table 1 to a figure (Figure 4 in the revised manuscript). We have now indicated the number of samples analyzed in the legend of the figure. (page 9, line 340)

8.    Please comment on whether pathological analysis was performed in a blinded manner?

Response: Yes, the pathological analysis was performed in a blinded manner by an independent veterinary pathologist (page 4, line 184)

9.    Figure 4Bi. It would be preferable, for consistency, if the authors chose a section in which 3 inlays were selected, as with all other panels in the figure.

Response: According to the veterinary pathologist, there are no other distinct histopathological changes can be spotted apart from bronchiolitis. As a result, we did not add additional two inlays in this figure. Figure 4 becomes Figure 5 in the revised manuscript. (Page 10, lines 359-398)

10. Figure 4&5 legends: How many animals is this data representative of?

Response: Three animals per group (n=3) were used for the histopathological analysis in these figures. Figures 4& 5 become Figures 5 and 6 in the revised manuscript. (Page 10, line 392; page 11, line 400)

Round 2

Reviewer 2 Report

Comments:

1. Suggestion for the future: please provide a clean revised manuscript with corresponding line numbers mentioned in the authors' reply. Current line numbers in the reply did not fully match with the marked up manuscript, which is difficult for the reviewer to go back and check.

2. Authors response to 4: How did the authors determine the LD50 in mice if they do not have a cell-line to measure virus titers (pfu/ml or TCID50)? To our knowledge, these two challenge viruses could be plaqued on MDCK cells.

3. Figure 5, it is understandable that the authors rotate the H&E images for 400 x magnification to show inflammations. But it is difficult to match up the two images of Fig 5B (i). Can the authors provide additional information to clarify that they are the same images? 

Author Response

1. Suggestion for the future: please provide a clean revised manuscript with corresponding line numbers mentioned in the authors' reply. Current line numbers in the reply did not fully match with the marked up manuscript, which is difficult for the reviewer to go back and check.

Response: We apologize for the inconvenience caused by the line numbers. We will ensure this will not happen again in future. Sometimes, however, we do notice that the line numbers might change when Words documents are saved as pdf files. The line numbers for the clean version of the manuscript are also different from the manuscript with track changes.

2. Authors response to 4: How did the authors determine the LD50 in mice if they do not have a cell-line to measure virus titers (pfu/ml or TCID50)? To our knowledge, these two challenge viruses could be plaqued on MDCK cells.

Response: The LD50 of these mouse-adapted viruses were determined using the Reed and Muench method described by Ramakrishnan (2016) (see Table 1 of the paper). The viruses that we purchased from ATCC stated that they had been adapted in eggs instead of MDCK cells. As a result, we did not plan to plaque the virus on MDCK cells. Nevertheless, we thank the reviewer for suggesting us to use MDCK cells. We will definitely consider to use this method in our future study.

Reference:

Ramakrishnan MA. 2016. Determination of 50% endpoint titer using a simple formula. World journal of virology 5:85-86. 10.5501/wjv.v5.i2.85 

3. Figure 5, it is understandable that the authors rotate the H&E images for 400 x magnification to show inflammations. But it is difficult to match up the two images of Fig 5B (i). Can the authors provide additional information to clarify that they are the same images?

Response: Million thanks for pointing out this mistake. We mistakenly inserted a wrong image into the old manuscript. We have now replaced the wrong figure with the image corresponds to the inlet of Fig 5Bi. We truly apologize for this mistake.

4. Moderate English editing required

Response: We have the revised manuscript checked and edited by Dr. Christopher Syme, University of Glasgow, UK. We have acknowledged Dr. Syme for his critical reading of the manuscript in the Acknowledgements section of the revised manuscript.  

Reviewer 3 Report

The authors have adequately addressed my prior concerns.

Author Response

Many thanks for your constructive comments.